# Antioxidant Properties of Oral Antithrombotic Therapies in Atherosclerotic Disease and Atrial Fibrillation

**DOI:** 10.3390/antiox12061185

**Published:** 2023-05-30

**Authors:** Luigi Falco, Viviana Tessitore, Giovanni Ciccarelli, Marco Malvezzi, Antonello D’Andrea, Egidio Imbalzano, Paolo Golino, Vincenzo Russo

**Affiliations:** 1Cardiology Unit, Department of Medical Translational Science, University of Campania “Luigi Vanvitelli”—Monaldi Hospital, 80126 Naples, Italy; luigifalco94@libero.it (L.F.); viviana.tessitore.vt@gmail.com (V.T.); ciccarelli.giovanni@gmail.com (G.C.); marcomalv2@hotmail.com (M.M.); paolo.golino@unicampania.it (P.G.); 2Cardiology Unit, Umberto I Hospital, 84014 Nocera Inferiore, Italy; antonellodandrea@libero.it; 3Department of Clinical and Experimental Medicine, University of Messina, 98122 Messina, Italy; egidio.imbalzano@unime.it

**Keywords:** thrombosis, anticoagulants, antiplatelet, antioxidant, oxidative stress

## Abstract

The thrombosis-related diseases are one of the leading causes of illness and death in the general population, and despite significant improvements in long-term survival due to remarkable advances in pharmacologic therapy, they continue to pose a tremendous burden on healthcare systems. The oxidative stress plays a role of pivotal importance in thrombosis pathophysiology. The anticoagulant and antiplatelet drugs commonly used in the management of thrombosis-related diseases show several pleiotropic effects, beyond the antithrombotic effects. The present review aims to describe the current evidence about the antioxidant effects of the oral antithrombotic therapies in patients with atherosclerotic disease and atrial fibrillation.

## 1. Introduction

Oxidative stress is due to the discrepancy between the rate of reactive oxygen species (ROS) generation and elimination. ROS are products of oxygen metabolism; and some environmental factors can trigger its production, tipping the scales toward an unbalanced state that results in cellular and tissue damage. Oxidative stress plays a key role in the onset of vascular thrombosis, both arterial and venous [1]. 

Oral antiplatelet therapies are the mainstay of the pharmacological treatment in patients with atherosclerotic disease since they have significantly reduced cardiovascular morbidity and mortality and achieved a class I recommendation both for coronary artery disease (CAD) [2,3,4] and symptomatic peripheral artery disease (PAD) [5].

Direct oral anticoagulants (DOACs) emerged as the preferred choice for the prevention of stroke and systemic embolism in patients with atrial fibrillation (AF) at increased thromboembolic risk; moreover, DOACs are indicated in the treatment of venous thromboembolism (VTE) [6,7,8]. DOACs achieved solid class I recommendations thanks to their comparable efficacy and better safety profile as compared to VKAs. These data derive both from several randomized clinical trials (RCTs) [9,10,11] and real-world observational studies of patient subgroups less represented in RCTs, such as severely obese patients [12,13], cancer patients [14,15], elderly patients [16,17,18,19,20,21], or patients who had undergone bioprosthetic valve replacement or prior repair [22,23,24]. 

Antithrombotic drugs have deeply modified the natural history of thrombosis-related diseases. Nevertheless, thrombotic disorders continue to pose a tremendous burden on healthcare systems, suggesting the need for a deeper understanding of thrombotic mechanisms. 

The translational research offers interesting pathophysiology tips, gradually shedding light on the pivotal role of oxidative stress in the genesis of cardiovascular diseases. In this context, both in vitro and in vivo studies have shown that antithrombotic drugs have antioxidant effects that could explain, at least in part, their favorable effects on clinical outcomes. 

The aim of the present review is to describe the role of oxidative stress in the thrombosis-related vascular disease and to summarize the main studies exploring the effects of oral antithrombotic drugs commonly used in patients with atherosclerotic disease and atrial fibrillation, on the oxidative stress pathways.

## 2. Role of Oxidative Stress in Thrombosis

Reactive oxygen species (ROS) derive from oxygen metabolism and involve either short-lived free radicals, oxygen ions, or more stable peroxides. Low levels of ROS are necessary in regulating many cellular functions and signaling pathways, working as second messengers, while excessive ROS production, together with exhaustion of endogenous antioxidant systems, provokes oxidative stress, deteriorating vascular cells’ functions through a wide range of mechanisms [25,26]. 

Therefore, oxidative stress is established when a mismatch between the production and clearance of ROS occurs. 

Several enzyme systems can potentially lead to ROS overproduction, such as NADPH oxidase (NOX), xanthine oxidase (XO), components of the mitochondrial respiratory chain, and endothelial nitric oxide synthase (NOS), when an uncoupling occurs, and to a lesser extent, cyclo-oxygenase (COX), lipoxygenase (LOX), and cytochrome p450 enzymes [27,28].

The leading generator of ROS in the vascular wall is the NOX system, which is found in endothelial cells (ECs), vascular smooth-muscle cells (VSMC), fibroblasts, monocytes, and macrophages infiltrating the inflamed vasculature. Unlike other enzymatic complexes that produce ROS because of their catalytic activity, NOXs are the only ones that primarily generate ROS [29]. Different isoforms of NOX exist and are variably expressed in diverse cell types, suggesting distinct functions. Several studies in animal models have elucidated the pivotal role of NOX isoforms in the development of vascular disorders. 

In apolipoprotein E-deficient (ApoE) mice, NOX subunit knockout decreased atherogenesis [30]. An antiatherogenic effect was observed in a similar model even when mice were fed an atherogenic diet or subjected to streptozotocin-induced diabetic mellitus [31]. 

Nox2 seems to be linked to the severity of atherosclerosis [32,33], Nox4 was demonstrated to have harmful effects in animal models of ischemic stroke [34], and Nox5 and p NOX subunits are overexpressed in human atherosclerotic lesions [35,36,37]. Finally, Vendrov et al. highlight that NOX is essential for low-density lipoprotein (LDL) oxidation [38]. ECs’ expression of XO is induced by biochemical triggers such as angiotensin II [39]. Additionally, a liver-derived circulating form of XO, generated by hypercholesterolemia, interacts with ECs through the interaction with glycosaminoglycans [40]. In atherosclerotic arteries, XO’s activity is increased and is responsible for the production of O_2_^−^ and H_2_O_2_ [37]; consequently, the use of XO inhibitors positively affects endothelial function and slows atherosclerosis progress in apoE-KO mice [41,42]. A healthy endothelium, via eNOS, generates NO, a crucial component of ECs’ vaso-protective mediators [43]. However, oxidative stress can impair the NOS function of producing NO. This promotes uncoupling of oxygen reduction and synthesis of NO [44]. Therefore, the ongoing oxidative stress drives eNOS to create superoxide at the cost of NO. Several models confirmed this mechanism as a main source of ROS [45,46,47]. Moreover, genetic iNOS deletion was found to minimize atherosclerosis in ApoE-KO mice in vivo [48]. However, most of the ROS derive from mitochondria [49].

Some experimental data suggest that, under pathological circumstances, mitochondrial dysregulation and ROS production may contribute to atherogenesis by stimulating the development of additional ROS and mitochondrial apoptotic signals [50,51]. Besides the mitochondrial respiratory chain, Mn superoxide dismutase (SOD) activity significantly accounts for ROS production; indeed, MnSOD deficiency was demonstrated to enhance the atherosclerotic process in ApoE/KO mice [52].

Oxidative stress and coagulation are closely intertwined. Indeed, ROS work at different levels in the coagulation landscape, involving endothelium, platelets, and coagulation factors [53]. One of the first consequences of an increased ROS production is endothelial dysfunction (ED). The hallmark of ED is the shift from a normal endothelium phenotype, that hampers platelets’ activation and fosters vasodilatation, towards an endothelium promoting a procoagulant state [54]. Additionally, platelets further boost oxidative stress, establishing a vicious circle that can hasten and strengthen the thrombotic process [53]. Mechanistic studies investigating molecular pathways underlying ROS effects on platelets’ functions are lacking. However, it has been hypothesized that the function of crucial receptors may be positively modulated by oxidative changes on sulfhydryl and thiol groups [55]. Therefore, coagulation factors’ generation and receptor-binding are boosted, and platelet adhesion is enhanced. Red blood cells (RBCs) are affected, too. RBCs, once thought to be passive spectators of hemostasis, are now recognized as crucial agents in fostering venous thrombosis and improving thrombus stability [56]. Oxidative stress can disrupt the membrane structure and foster phosphatidylserine exposition, laying the groundwork for prothrombin cleavage [57]. In addition, RBCs’ mechanical properties are impaired, and thus blood viscosity is enhanced. Subsequently, local blood flow is lowered, enabling RBCs’ aggregation, and supporting platelet adhesion [58].

### 2.1. Role of Oxidative Stress in Atherogenesis

The storage of low-density lipoproteins (LDL) in the intima is strongly related to the development of the fatty streak [59], which is the first morphological alteration that occurs in atherosclerosis [60]. In this context, LDL may become more susceptible to oxidation, which favors phagocyte absorption. 

LDL oxidation, indeed, is implicated in the progression of atherosclerosis in several ways. First, while LDL internalization is counteracted by the receptor downregulation, oxidized LDLs (oxLDLs) are extensively taken-up via scavenger receptors, resulting in significant cholesterol accumulation, which leads to the generation of foam cells [61].

The modified LDLs trigger the endothelial response either directly, stimulating the release of mediators, or indirectly, through the activation of macrophages, which start an immunological response by identifying altered LDLs as foreign substances, feeding the inflammatory response [62,63]. 

This process causes expression of adhesion molecules (such as vascular cell adhesion molecule-1 (VCAM-1) and intercellular adhesion molecule-1 (ICAM-1)), growth factors, and cytokines in endothelial cells and VSMCs, promoting extravasation, differentiation, and polarization of macrophages [64,65]. A proinflammatory M1-like phenotype conversion occurs in infiltrated macrophages; therefore, they contribute to vascular wall release of proinflammatory factors, lytic enzymes, and ROS, perpetuating a feedback loop [66].

However, several other processes that underlie atherogenesis and endothelial dysfunction are influenced by oxidative stress. ROS impair endothelial permeability, hindering vascular barrier function [67], and increase transcription of nuclear transcription factor-κβ (NF-kB), fostering proinflammatory genes’ expression and disrupting the junctional proteins network [68,69,70]. Finally, another key regulator of the endothelial response suffering from increased ROS generation is glycocalyx [71]. ROS enhances the activity of metalloproteinase, leading to the deterioration of proteoglycans’ structure, while oxLDLs lower its thickness [72,73]. This results in augmented permeability, fostering further lipid retention [74] (Figure 1).

### 2.2. Role of Oxidative Stress in Venous Thrombosis

Oxidative stress seems to be implicated in venous thrombosis since some evidence suggests an increased tissue factor (TF) expression in ECs and VSMCs after ROS exposition [75,76,77]. 

ROS produced by the mitochondrial respiratory chain and NOX induced a proinflammatory and procoagulant state in experimental models [78,79]. Moreover, as a feedback mechanism, thrombin was found able to further elicit ROS production, worsening ED [80]. 

The oxidative changes of proteins involved in the modulation of the hemostatic process have been proposed as an additional mechanism. The tissue factor pathway inhibitor (TFPI) is the main physiological regulator of TF activity and may be blocked by oxidative stress, impacting the thrombotic process [81]. ROS may directly inactivate both the protein C and its upstream agonist thrombomodulin [82,83]. Conversely, other studies suggested an indirect inhibition preventing the binding between these endogenous anticoagulants and thrombin [84]. Finally, fibrinogen, once oxidized, is more easily cleaved into fibrin [85] (Figure 2).

### 2.3. Role of Oxidative Stress in Atrial Fibrillation

Atrial fibrillation (AF) is the most common sustained arrhythmia [6] and the major cause of cardioembolic stroke. Rising prevalence of AF has gradually paved the way for a shift in stroke etiology [8]; therefore, the cardioembolic stroke is now the most frequent type of ischemic stroke [8].

AF shares the same risk factors as atherosclerosis [86], and its prevalence increases with aging [87]. Therefore, it has been hypothesized that oxidative stress contributes to the pathophysiology of AF [88,89]. Indeed, previous models have shown enhanced oxidation of myocardial proteins and oxidative damage of atrial tissue [90,91]. 

The impaired intracellular Ca^2+^ regulation plays a key role in the modification of electrical homeostasis. AF-derived atrial myocytes exhibit enhanced diastolic sarcoplasmic reticulum (SR) Ca^2+^ leak through the ryanodine receptor (RyR2) [92,93]. In addition, a mutation-induced intracellular Ca^2+^ leak may trigger AF in knock-in mice [94,95]. Some data suggest that RyR2 may be a target of oxidized calmodulin-dependent protein kinase II (CaMKII), that phosphorylates the receptor, leading to a Ca^2+^ leak [96]. Xie et al. [97] showed that RyR2 is oxidized by mitochondrial-produced ROS in atrial myocytes, increasing the intracellular Ca^2+^ leak. Interestingly, the decrease of ROS levels limited the Ca^2+^ leak and stopped AF.

## 3. Antioxidant Effects of Antiplatelet Drugs

### 3.1. Aspirin 

The antioxidant properties of aspirin are huge and extremely heterogeneous, since it seems to act at different levels. 

Several preclinical studies [98,99,100] have showed the antioxidant effects of aspirin in bovine pulmonary artery endothelial cells and human umbilical vein endothelial cells (HUVECs). Aspirin exerted direct endothelial protection, mitigating H_2_O_2_-induced toxicity [98] and stimulating the ferritin synthesis [100,101,102]. These effects occurred at therapeutically relevant concentrations and after several hours of pretreatment, suggesting a triggered gene transcription as a possible mechanism. Moreover, aspirin enhances both the expression and activity of heme-oxygenase-1 (HO-1), which provides a strong defense against oxidative tissue injury through several mechanisms. One of these is enhancing bilirubin and carbon monoxide, which exert an antioxidant activity. HO-1 involvement was shown in either in vitro or in vivo models [103,104,105,106,107]. Several preclinical studies added details on aspirin-enhanced HO-1 activity. The Nrf2-ARE pathway plays a critical role in upregulating the antioxidant and detoxifying enzymes in the body, with HO-1 being the main downstream product [108].

Aspirin leads to an antioxidant property also through inhibition of NF-kB. NF-kB is a key modulator of inflammatory reactions and affects many different facets of both the innate and adaptive immune systems. NF-kB is involved in the control of the inflammasome and the induction of the expression of several proinflammatory genes, such as those encoding cytokines and chemokines [109].

Reduction of NF-kB transcription was observed in many in vitro [110,111] and in vivo models [112]. The final effect was a reduction of ROS levels and proinflammatory cytokines. 

Yang et al. explored a novel mechanism by which aspirin limits NF-kB effects. They demonstrated that aspirin increases activator protein 2α (AP-2α) phosphorylation, upregulating the inhibitor of nuclear factor kappa B (IkBα). This effect was found both in apoE-/-mice and in humans and resulted in a reduction of oxidative stress and of the plaque instability [112]. 

The antioxidant implication of aspirin seems to also be related to the reduction of lipid peroxidation, and this was a common finding in many investigations, in vitro [113,114,115,116] and in vivo [117,118,119].

The balance between pro-oxidant and antioxidant systems is complex and aspirin plays a crucial role, interfering with activities of different enzymatic systems. Wròbel et al. [120] examined redox homeostasis in the liver and brain of BALB/c mice after aspirin delivery. There was an enhanced activity of 3-mercaptopyruvate sulfur transferase and g-cystathionase, i.e., enzymes involved in the production of H_2_S, and the GSH/GSSG ratio was higher, thus reflecting a higher antioxidant capacity [120]. 

In many experimental models [112,121,122,123], aspirin exerts a useful function, suppressing NOX activity, which augmented oxidative stress, producing superoxide ion, at the expense of NADPH.

In other in vivo models, aspirin improved glutathione peroxidase (GP), glutathione transferase (GT) [124] and CAT function [125,126,127]. CAT is an antioxidant enzyme which converts H_2_O_2_ into water and oxygen, reducing the toxicity related to ROS production. The antioxidant property was also related to the ability of aspirin to enhance the transcription of genes encoding for these antioxidant enzymes (CAT, SOD) [126]. In two models of ischemia-reperfusion injury administration, aspirin led to a significant reduction in nitro-oxidative stress and an improvement of endothelial function [128,129]. Moreover, in one study, aspirin antioxidant properties protected from oxidative stress-induced DNA strand breaks [130].

Four studies evaluated the antioxidant effects of aspirin in humans [131,132,133,134], and two of them were conducted in healthy middle-aged subjects. 

In a pilot study, Ristimae et al. [131] evaluated the serum products of lipid peroxidation in 25 healthy men at baseline and after two weeks of treatment with 100 mg of aspirin. No significant modifications were observable in SOD and CAT activity nor in GSH levels; however, in the aspirin group, a higher serum antioxidant capacity (AOC), expressed by the ability of serum to inhibit peroxidation of linoleic acid, has been shown. 

Kurban et al. [132] confirmed this preliminary observation among 30 healthy volunteers receiving 100 or 150 mg of aspirin. After two months of treatment, patients on 150 mg of aspirin showed a significantly lower total oxidative status (TOS) and serum oxLDLs levels. These results suggest that ASA treatment may contribute to the prevention of atherosclerosis, a beneficial effect which is dose- and time-dependent.

Cheng et al. [134] evaluated the effects of aspirin in two clinical scenarios: patients with stable angina (SA) and those referred for a first-time coronary artery bypass graft (CABG). 

Among the patients with stable angina, 100 mg of aspirin showed a significant decrease in circulating microparticles (MPs) levels, which derive from either activated platelets or endothelial vascular cells and promote coagulation and ED [134]. Among the patients referred for a first-time coronary artery bypass graft (CABG), 160 mg of aspirin until one day before the heart surgery reduced the oxidative stress and inflammation, significantly lowering the serum levels of 8-iso-PGF2α and high-sensitivity C-reactive protein (hs-CRP) [133] (Table 1).

### 3.2. P2Y12 Inhibitors

The interaction between the P2Y12 receptor and adenosine diphosphate (ADP) is essential for thrombus formation, leading to the release of platelet-dense granules, the inhibition of the intracellular prostacyclin pathway, and finally, the promotion of platelet aggregation. Thus, P2Y12 has become an attractive target for the latest generation of antiplatelet therapies [135]. The potential antioxidative property of P2Y12 inhibitors has so far been evaluated mainly with clopidogrel. Clopidogrel is a second-generation thienopyridine that irreversibly binds P2Y12, inhibiting platelet aggregation. As an inactive prodrug, clopidogrel needs to undergo hepatic bioactivation. However, this step involves only 15% of the administered prodrug, as the remaining 85% is extensively converted into a non-functional metabolite [136]. Despite the pharmacokinetic profile and the rise of more potent third-generation P2Y12 inhibitors (prasugrel and ticagrelor), clopidogrel is still widely prescribed [137,138]. Some preclinical studies show that clopidogrel diminishes lipid peroxidation, and therefore the MDA level, as a marker of oxidative stress damage. This effect was observed in several in vivo models [139,140,141]. Clopidogrel was able to not only reduce ROS toxicity but also to prevent a GSH decrease, enhancing natural and endogenous antioxidant systems. In two studies, clopidogrel was evaluated in the context of ischemia reperfusion injury models [139,140]. Treatment with clopidogrel reduced oxidative stress toxicity. In BALB-c mice, clopidogrel administration resulted in a higher total antioxidant capacity, despite the unchanged levels of CAT, SOD, and GSH, and it reduced renal cell apoptosis [140].

In human aortic endothelial cells (HAECs), clopidogrel has proven to have a vaso-protective action: it counteracts oxidative stress through the activation of CaMKK𝛽/AMPK/Nrf2 pathways, and by increasing GSH and HO-1 expression [142]. 

Clopidogrel also exerted anti-inflammatory and antioxidant functions in wild-type mice infused with angiotensin II. Angiotensin II causes vascular dysfunction and platelet activation, enhancing the inflammation response. Clopidogrel was shown to reduce platelet deposition and the platelets–monocytes interaction induced by angiotensin [143]. 

In another study, sixty male mice fed a high-fat diet were evaluated after isoproterenol-induced myocardial ischemia. Pretreatment for eight weeks was performed either with clopidogrel or aspirin. Mice in the clopidogrel group had better protection against myocardial infarction. Indeed, as compared with the aspirin group, metalloproteinases, hsCPR, and troponin C were further reduced. Conversely, CAT activity and the redox state improved [144]. 

Three studies assessed ticagrelor’s antioxidant properties [145,146,147]. Kang et al. [145] induced apoptosis in HUVECs through oxLDLs administration. Ticagrelor diminished apoptosis in a dose-dependent fashion, proving protection against oxidative damage. In the study of El-Mokadem et al. [146], ticagrelor was evaluated in renal ischemia-reperfusion injury. Ischemia was induced in Wistar rats by clamping renal arteries. Animals in the treatment group received ticagrelor for three days after reperfusion, showing a significant reduction of lipoperoxidation, TNF-alpha, and NF-kB expression [146]. Finally, Bitirim et al. [147] showed that the extracellular vesicles, derived from ticagrelor-exposed high-glucose-incubated H9c2-cells (H9c2), dramatically reduced ROS generation; in addition, the oxidative stress-associated miRNA expression profile was mitigated. Grzesk et al. [148] demonstrated that among the P2Y12 inhibitors, only ticagrelor avoided the ADP-induced VSMC contraction. This effect, derived from the interaction with extra-platelet-located P2Y12 receptors, enriches the evidence of pleiotropic effects of this new potent antiplatelet drug. 

Several in-human studies support the hypothesis of the pleiotropic effects of the P2Y12 inhibitors.

P2Y12 inhibitors are, in fact, the most investigated drugs in randomized studies among antithrombotic medications [149,150,151,152,153]. 

Heitzer et al. [149] evaluated daily clopidogrel administration after a 300 mg loading dose in patients with stable coronary artery disease (CAD) already receiving aspirin. Participants were randomized to take a P2Y12 inhibitor or a placebo. After five weeks, patients receiving clopidogrel had lower levels of urinary 8-iso-PG F2α and sCD40L, reducing proinflammatory stimuli. Clopidogrel ameliorates ED, thus improving NO bioavailability.

Taher et al. [150] documented a rise in GSH levels in diabetic patients randomized to clopidogrel therapy without a loading dose. Uncontrolled diabetic patients were randomized into two groups: clopidogrel and placebo groups, both already receiving hypoglycemic therapy. After two months, a reduction of oxidative stress as well as a reduction of glycemic parameters was observed in the clopidogrel group. 

Circulating endothelial cells (CECs) are an established marker of vascular injury in people with diabetes [154]. Treatment with clopidogrel seems to reduce the number of CECs. Participants of the study received clopidogrel at 75 mg for 30 days. Blood specimens were collected before and after the treatment. CECs returned to almost normal levels; moreover, clopidogrel increases the level of phosphorylated Akt and phosphorylated adenosine monophosphate kinase, which increases the Enos activity and provides a beneficial endothelial function [155]. 

Another observational study evaluates the antioxidant effects of clopidogrel in patients undergoing percutaneous coronary intervention (PCI). Bundhoo et al. examined 58 patients with SA. All of them underwent PCI with stent implantation and received aspirin before the procedure. One group received clopidogrel at a 600 mg loading dose, while the second group was already on a clopidogrel maintenance dose of 75 mg for at least three days. Both loaded patients and chronic therapy patients showed a beneficial trend of the plasma total antioxidant capacity (TAC) [156]. 

More recently, the antioxidant effects of two different P2Y12 strategies were evaluated in stable CAD patients requiring PCI [152,153]. 

The use of ticagrelor on top of aspirin, for at least one month, in CAD patients with chronic pulmonary obstructive disease (COPD) showed a significant reduction in the apoptosis rate and ROS production when patients’ serum was added to HUVECs [152]; moreover, an increased expression of sirtuin1 (SIRT1) and hairy enhancer of split-1 transcription factor (HES-1), two antioxidant transcription factors, was shown [153]. 

The antioxidant effect of prasugrel has been evaluated in a single, randomized, active-controlled study including patients with unstable angina in need of PCI. After three months of 10 mg of prasugrel, a significant reduction in MPO levels, sCD40L, and nitrite levels was shown [151] (Table 2).

## 4. Antioxidant Effects of DOACs

### 4.1. Rivaroxaban

The antioxidant properties of rivaroxaban have been investigated in some experimental [157,158,159,160,161,162,163,164,165] and animal models [166,167,168,169] (Table 3).

Three preclinical studies employed HUVECs as experimental models [158,161,163]. The exposition of human plasma and advanced glycation end products to HUVECs led to ROS production and adhesion molecules’ expression. These effects were reverted by rivaroxaban in a dose-dependent fashion [158]. Reduced levels of adhesion molecules, oxidative stress biomarkers, proinflammatory cytokines, and chemokines were also demonstrated in the other two HUVECs models [161,163]. Rivaroxaban seems to mediate these functions through inhibition of the PAR pathway, activated by thrombin signaling.

Ishibashi et al. [159] tested rivaroxaban on human proximal tubular cells treated with factor Xa. Rivaroxaban significantly decreased ROS and MCP-1. 

Most preclinical studies, both in vitro and in vivo, were conducted on Wistar rats [160,162,164,165,167,168,169].

Gul Utku et al. demonstrated the healing effects of rivaroxaban in a model of induced colitis due to its favorable activity on MPO and SOD [160]. In two models of liver fibrosis, Vilaseca et al. showed that rivaroxaban improved the ED and reduced oxidative stress [162]. Rivaroxaban also exerted a protective effect in a model of testicular injury, where rats pretreated with rivaroxaban maintained higher levels of endogenous antioxidants and showed lower expression of NF-kB [164]. Rivaroxaban also mitigates the lipid peroxidation, as demonstrated by the lower levels of MDA in an ischemia-reperfusion model of Sprague-Dawley rats [157].

Wistar rats were the preferred model for in vivo studies as well [167,168,169]. Two studies used sunitinib, an oral tyrosin kinase inhibitor, to induce oxidative stress, resulting in renal injury and cardiotoxicity. Intraperitoneal administration of sunitinib resulted in renal injury [168] and cardiotoxicity [167]. Rivaroxaban-treated rats exhibited lower levels of MDA and TNF-alpha; conversely, GSH and glutathione reductase decreases were aborted. Moreover, cardiac fibrosis and remodeling were mitigated due to diminished NF-kB expression and ROS production [166]. Rivaroxaban was also able to reduce oxidative stress in brain tissue, in an experimental model of depression [169].

A unique human ex vivo investigation evaluated the influence of rivaroxaban on abdominal aortic aneurysmal sites with intraluminal mural thrombus. In the rivaroxaban pretreated group, a decreased expression of ICAM-1, VCAM-1, and IL-6 was reported. Moreover, levels of NOX subunits were reduced [170].

In the COMPASS trial [171], the combination of 100 mg of aspirin and 2.5 mg of rivaroxaban twice daily significantly reduced the cardiovascular death, stroke, and myocardial infarction in patients with stable CAD and/or peripheral artery disease (PAD). These positive results led to several studies investigating the synergistic effects of antiplatelet and anticoagulant medications. Abedalqader et al. [165] examined rivaroxaban and aspirin effects in a model of isoproterenol-induced cardiac injury. Combination therapy was associated with a decrease in TBARS and IL-6 levels. However, no significant difference was observed between combination therapy and rivaroxaban, or aspirin administered as a monotherapy. Recently, Russo et al. [172] showed that dual-pathway inhibition with low-dose rivaroxaban and aspirin in patients with an established diagnosis of CAD and/or PAD was associated with a reduction in serum levels of some inflammation markers, such as IL-6 and fibrinogen. Moreover, this combined therapy showed little to no impact on hemoglobin values and renal function markers. These findings support the hypothesis of a pleiotropic anti-inflammatory effect of rivaroxaban, in addition to its anticoagulant effect, and partially explain the positive results of the COMPASS trial for the reduction of cardiovascular events in patients with stable atherosclerotic vascular disease.

### 4.2. Apixaban 

Few data are available about the antioxidant effects of apixaban. In a preclinical in vitro study, Torramade-Moix et al. [173] showed that the preincubation of HUVECs and human dermal microvascular endothelial cells (HMECs-1) with apixaban was associated with normalization of ROS levels, increasing eNOS expression, and the reduction of adhesion molecules’ expression (Table 4).

### 4.3. Edoxaban

Two preclinical studies described the role of edoxaban on oxidative stress [174,175].

Edoxaban antioxidant activity was assessed in a model of human proximal tubular cells (human kidney 2 cells (HK-2 cells)) exposed to several oxidative stress stimulants. Edoxaban blunted ROS production induced by angiotensin II, indoxyl-sulfate, and factor Xa; moreover, it was able to effectively scavenge peroxynitrite and O2-. Notably, edoxaban scavenging activity was independent from its action on factor Xa, but it was derived from its peculiar molecular structure [174]. 

In a recent study by Fang et al. [175], wild-type mice underwent a subtotal nephrectomy at 8 weeks of age and were randomized to receive edoxaban or a normal diet. Edoxaban-treated mice showed lower levels of inflammatory and oxidative stress biomarkers; moreover, the HK-2 cells’ preincubation with edoxaban resulted in a lower expression of NF-kB (Table 5).

### 4.4. Dabigatran

Two early preclinical studies [176,177], including ApoE−/−mice fed a high-fat diet, showed a considerable reduction in the mean plaque area and an increased thickness of the fibrous cap, in those randomized to receive dabigatran. These modifications were linked to lower NF-kB levels. 

In the mouse models of neurodegenerative diseases, dabigatran significantly reduced the ROS levels and the expression of iNOS and NOX [178,179,180].

In ischemia-reperfusion models [181,182], dabigatran, started 1 week before ischemia induction, significantly decreased lipoperoxidation compared to LWMH [181], and showed a similar effect in reducing TOS and proinflammatory cytokines compared to apixaban and rivaroxaban [182]. Dabigatran provided complete protection against DNA strand breakage [183].

Treatment with dabigatran significantly inhibited the P65 of nuclear factor κB, tumor necrosis factor α, interleukin (IL)-1β, and IL-6 activities, and significantly enhanced the catalase and superoxide dismutase activities in the AMI rabbits [184].

The antioxidant effects of dabigatran were confirmed in models of CCl4-induced liver fibrosis [185], arthritis in Wistar albino rats [186], nephrotoxicity induced by cisplatin [187], and tubulointerstitial fibrosis obtained with ureteral obstruction [188] (Table 6).

## 5. Clinical Implications

The beneficial effects of aspirin, clopidogrel, ticagrelor, and rivaroxaban on the oxidative stress system have been shown in preliminary observational clinical studies including patients with CAD; however, the relationship between the antioxidant properties and the occurrence of cardiovascular events has not been demonstrated. The choice of the oral antithrombotic therapy based on the need of its antioxidant properties should follow a patient-centered approach. In the future, the use of oxidative stress biomarkers could help to identify these patients. The long-term management of antiplatelet and anticoagulant regimens may be complex in real-world settings, especially in elderly patients with comorbidities [189,190,191,192]. Bleeding risk remains a major issue, as clinicians have to carefully monitor the changing risk over time and take into account the residual thrombotic risk. In the setting of acute coronary syndrome and a high bleeding risk, the use of the P2Y12 inhibitor or aspirin monotherapy after a short period of dual-antiplatelet therapy (DAPT), or the early shift from newer P2Y12 inhibitors to clopidogrel, are considered potential strategies to reduce the bleeding risk [192,193]. DOACs emerged as the preferred choice for the prevention of stroke and systemic embolism in AF patients at an increased thromboembolic risk; moreover, DOACs are indicated in the treatment of VTE. Even if much lower than VKAs, some drug–food and drug–drug interactions have been described during DOACs treatment due to the interaction with P450 cytochromes, ABC transporters, and P-glycoprotein (P-gp) [194,195,196]. When starting a DOAC, knowledge of current kidney and liver functions is required. Importantly, kidney function should be assessed using the Cockcroft–Gault formula, as it was used in the pivotal phase III RCTs. Moreover, a baseline hematological profile should be obtained for reference during long-term follow-up. [195] The use of DOACs in daily clinical practice does not require monitoring of coagulation since all seminal RCTs comparing DOACs to VKAs have been conducted without dose adjustments based on plasma level measurements. However, conflicting results have emerged from the analysis of RCTs and long-term longitudinal studies [197,198,199,200]. Therefore, assessment of the anticoagulant effect of DOACs may be desirable in certain rare situations, such as extreme body weight, concomitant oncologic therapies, patients after transplantation, patients on HIV medication, etc. [201,202].

## 6. Conclusions

The oxidative stress plays a role of pivotal importance in the physiopathology of CAD, peripheral artery disease, venous thrombosis, and atrial fibrillation. Several experimental in vitro and animal studies have shown the pleiotropic antioxidant effects of both oral antiplatelet and anticoagulant therapies. In the clinical setting, the beneficial effects of aspirin, clopidogrel, ticagrelor, and rivaroxaban on the oxidative stress system have been demonstrated by preliminary observational clinical studies including patients with coronary artery disease. Whether this mechanistic evidence may have an impact on lowering cardiovascular events is an intriguing question to investigate.

## Figures and Tables

**Figure 1 antioxidants-12-01185-f001:**
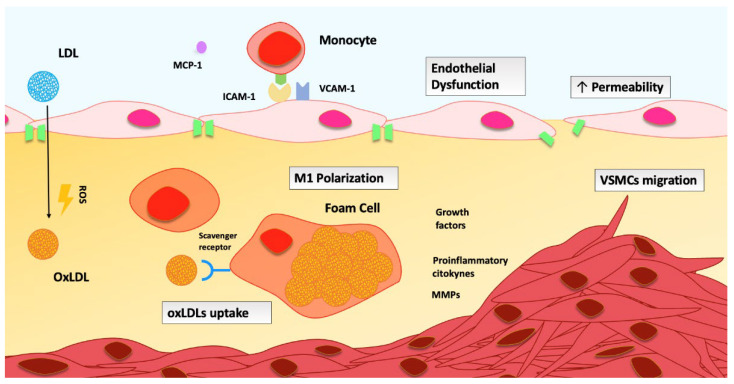
**Role of ROS in atherogenesis.** LDLs: low-density lipoproteins; OxLDLs: oxidized low-density lipoproteins; ROS: reactive oxygen species; MCP-1: monocyte chemoattractant protein-1; ICAM-1: intercellular adhesion molecule-1; VCAM-1: vascular cell adhesion molecule-1; MMPs: metalloproteinases; VSMCs: vascular smooth-muscle cells.

**Figure 2 antioxidants-12-01185-f002:**
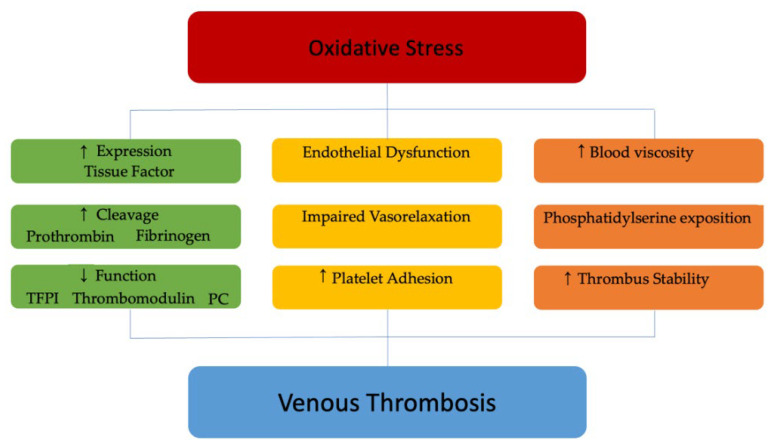
**Role of ROS in venous thrombosis.** TFPI: tissue factor pathway inhibitor; PC: protein C.

**Table 1 antioxidants-12-01185-t001:** **Main studies evaluating the antioxidant effects of aspirin**. HUVECs: human umbilical vein endothelial cells; HO-1: heme-oxygenase 1; NOX: NADPH oxidase; ROS: reactive oxygen species; NRF-2: nuclear factor E2-related factor 2; ARE: antioxidant response element; CAT: catalase; PARP: poly(ADP) ribose polymerase; HSP70: heat stress proteins 70; NF-kB: nuclear transcription factor-kB; SOD: superoxide dismutase; iNOS: inducible nitric oxide synthase; GSH: reduced glutathione; GSSG: oxidized glutathione; GP: glutathione peroxidase; GT: glutathione transferase; H_2_S: hydrogen sulfide; IkBα: nuclear transcription factor-kB inhibitor alpha; TBARS: thiobarbituric acid reactive substances; TOS: total oxidative status; TAC: total antioxidant capacity; OSI: oxidative stress index; MDA: malondialdehyde; TNFa: tumor necrosis factor a; oxLDLs: oxidized low-density lipoproteins; ApoE: apolipoprotein E; AS: stable angina ; CABG: coronary artery bypass graft.

Authors	Protocol	Patients	Target	Aspirin Dose	Results
Podhaisky [98]	Preclinical in vitro	-	Bovine pulmonary artery endothelial cells	-	↓ H202-induced toxicity ↑ cell viability
Oberle [99]	Preclinical in vitro	-	Bovine pulmonary artery endothelial cells	-	↑ Ferritin protein synthesis
Shi [113]	Preclinical in vitro	-	Mouse macrophage cells	-	↓ silica-induced lipid peroxidation ↓ DNA strand breakage ↓ activation of NF-kB
Hsu [130]	Preclinical in vitro	-	ϕ-174 plasmid DNA	-	↓ Oxidative stress-induced DNA damage
Grosser [100]	Preclinical in vitro	-	HUVECs	-	↓ H202-mediated toxicity ↑ HO-1 protein expression
Wang [123]	Preclinical in vitro	-	ventral mesencephalic tissues of embryonic day 14–15 rats	-	↓ NOX activity ↓ superoxide production by microglial cells ↓ intracellular ROS concentrations
Jian [104]	Preclinical in vitro	-	human primary melanocytes	-	↑ cell viability intracellular ↓ ROS levels ↑ NRF2 nuclear translocation ↑ ARE-luciferase activity ↑ expression of HO-1
Dimitrovska [127]	Preclinical in vitro	-	Adult male Wistar rats	-	↑ CAT activity ↑ PARP and HSP70
Wang [105]	Preclinical in vitro	-	Adult male Sprague-Dawley rats	-	↑ Nrf2/HO-1 signaling pathway
Chen [110]	Preclinical in vitro	-	NF-kB–luciferase+/+transgenic mice	-	↓ NF-kB expression ↓ ROS
Jorda [111]	Preclinical in vitro	-	Wilson rats	-	↑ cell viability ↓ Cu/Zn-SOD and Mn-SOD ↓ NF-kB expression ↓ iNOS
John [115]	Preclinical in vitro	-	Goto-Kakizaki Diabetic Rats	-	↓ ROS ↓ NOX activity ↓ lipid peroxidation ↓ CYP2E1 activity
John [116]	Preclinical in vitro	-	Goto-Kakizaki Diabetic Rats	-	↓ Kidney ROS ↓ NOX activity ↓ lipid peroxidation ↑ SOD activity ↑ GSH
Veres [128]	Preclinical in vitro	-	Lewis rats	-	↓ nitro-oxidative stress
De La Cruz [124]	Preclinical in vitro and ex vivo	-	Adult male Wistar rats	-	↓ oxidative stress, ↓ iNOS activty ↓ TBARS formation, ↓ GSSG ↑ GP and GT activity
Wròbel [120]	Preclinical in vitro and ex vivo	-	male BALB/c mice	-	↑ H_2_S-producing enzymes activity ↑ GSH/GSSG ratio
Wu [121]	Preclinical in vitro and in vivo	-	Male Sprague-Dawley, Wistar-Kyoto, and spontaneously hypertensive rats	-	↓ O_2_ production ↓ NOX activity
Yang [112]	Preclinical in vitro and in vivo	-	ApoE−/−mice, vascular smooth-muscle cells	-	↑ IkBα ↓ NOX activity ↓ F2-isoprostanes
Frydrychowski [129]	Preclinical in vitro and in vivo	-	Female pigs	-	↓ TOS, OSI, MDA ↑ TAC
Caballero [117]	Preclinical in vivo	-	Streptozotocin-induced diabetic male CF1 mice	-	↓ accumulation of lipoperoxidative aldehydes
De Cristòbal [118]	Preclinical in vivo	-	Adult male Wistar rats	-	↓ iNOS activity and expression ↓ TNF-a ↓ lipid peroxidation and concentration of GSSG
Prasad [114]	Preclinical in vivo	-	New Zealand white female rabbits	-	↓ serum and aortic MDA ↓ white blood cell-chemiluminescence ↑ antioxidant reserve
Kiliçoğlu Aydin [125]	Preclinical in vivo	-	Spraque-Dawley rats	-	↑ Erythrocyte CAT ↓ Erythrocyte SOD
Ayyadevara [126]	Preclinical in vivo	-	Caenorhabditis elegans	-	↑ Transcript levels of antioxidant genes encoding SOD, CAT, GT ↑ resistance to exogenous peroxide ↓ ROS
Chaávez [119]	Preclinical in vivo and ex vivo	-	Wistar male rats	-	↓ Lipid peroxidation ↑ GSH/GSSG ↑ GSH ↓ NF-kB
Wu [122]	Preclinical in vivo and ex vivo	-	Cardiomyopathic male hamsters	-	↓ O_2_^−^ production ↓ NOX activity
Ristimäe [131]	Non-blind, non-placebo-controlled study	25	Healthy middle-aged subjects	100 mg	↑ serum antioxidative capacity
Kurban [132]	Non-blind, non-placebo-controlled study	30	Healthy middle-aged subjects	100–150 mg	(150 mg group): ↓ TOS ↓ OxLDLs
Cheng [134]	Non-blind, non-placebo-controlled study	80	50 AS and 30 healthy middle-aged subjects	100 mg	↓ O_2_^−^ ↓ NF-kB
Berg [133]	Prospective, randomized	20	Patients referred for first-time CABG	160 mg	↓ 8-iso-PGF2α

**Table 2 antioxidants-12-01185-t002:** **Main studies evaluating the antioxidant effects of P2Y12 inhibitors**. MDA: malondialdehyde; GSH: reduced glutathione; SOD: superoxide dismutase; TAC: total antioxidant capacity; TNFa: tumor necrosis factor a; HO-1: heme-oxygenase 1; ROS: reactive oxygen species; NOX: NADPH oxidase; CAT: catalase; oxLDL: oxidized low-density lipoproteins; CECs: circulating endothelial cells; EPCs: endothelial progenitor cells; SIRT1: sirtuin1; HES1: hairy and enhancer of split-1; MPO: myeloperoxidase; HUVECs: human umbilical vein endothelial cells; HAECs: human amniotic epithelial cells; DMII: type II diabetes mellitus; COPD: chronic obstructive pulmonary disease; CAD: coronary artery disease; PCI: percutaneous coronary intervention; ASA: acetylsalicylic acid (aspirin).

Authors	Protocol	Patients	Target	P2Y12 Drug and Dose	Results
Kanko [139]	Preclinical in vitro and in vivo	-	Male adult Sprague-Dawley rats	Clopidogrel	↓ MDA ↑ GSH ↑ SOD activity
Hu [140]	Preclinical in vitro and in vivo	-	Healthy male BALB/c mice	Clopidogrel	↓ apoptosis ↑ TAC
Hadi [141]	Preclinical in vitro	-	Domestic rabbits	Clopidogrel	↓ MDA ↑ GSH
Yang [142]	Preclinical in vitro	-	HAECs	Clopidogrel	↓ TNF-a ↓ ROS ↑ GSH ↑ HO-1
An [143]	Preclinical in vitro	-	Wild-type male mice	Clopidogrel	↓ NADPH
Korish [144]	Preclinical in vitro	-	Male mice	Clopidogrel	↓ MDA ↑ catalase activity
Kang [145]	Preclinical in vitro	-	HUVECs	Ticagrelor	↓ oxLDL-induced apoptosis
El-Mokadem [146]	Preclinical in vitro	-	Adults male Wistar rats	Ticagrelor	↓ MDA ↓ TNF-a ↓ apoptosis
Bitirim [147]	Preclinical in vitro	-	High-glucose-incubated H9c2-cells	Ticagrelor	↓ ROS ↓ apoptosis ↑ miR-499, miR-133a, miR-133b
McClung [155]	Non-blind, non-placebo-controlled study	9	Patients with DMII	Clopidogrel 75 mg	↓ CECs ↑ EPCs
Bundhoo [156]	Non-blind, non-placebo prospective study	58	Patients undergoing PCI	Clopidogrel loading dose of 600 mg, then 75 mg od	↑ TAC
Heitzer [149]	Prospective, randomized	103	Patients with stable CAD and chronic ASA therapy	Clopidogrel loading dose of 300 mg, then 75 mg od	↓ urinary 8-iso-PG F2α
Taher [150]	Prospective, randomized	42	Patients with DMII	Clopidogrel 75 mg	↓ MDA ↑ GSH
Campo [152]	Prospective, randomized	46	Patients with COPD and stable CAD requiring PCI	Clopidogrel loading dose if 300 mg, then 75 mg od, or ticagrelor loading dose of 180 mg, then 90 mg bid	↓ ROS (ticagrelor group)
Aquila [153]	Prospective, randomized	46	Patients with COPD and stable CAD requiring PCI	Clopidogrel loading dose of 300 mg, then 75 mg od, or ticagrelor loading dose of 180 mg, then 90 mg bid	↑ expression of SIRT1 and HES1
Rudolph [151]	Randomized, active-controlled, double-blind trial	45	Patients undergoing PCI	Clopidogrel loading dose of 600 mg, then clopidogrel 75 mg od, or prasugrel 10 mg od	↓ MPO

**Table 3 antioxidants-12-01185-t003:** **Main studies evaluating the antioxidant effects of rivaroxaban**. MDA: malondialdehyde; ROS: reactive oxygen species; MCP-1: monocyte chemoattractant protein-1; ICAM-1: intercellular adhesion molecule 1; VCAM-1: vascular cell adhesion protein 1; MPO: myeloperoxidase; NOX: NADPH oxidase; NOS: nitric oxide synthase; SOD: superoxide dismutase; GP: glutathione peroxidase; NF-kB: nuclear transcription factor-κB; TBARS: thiobarbituric acid reactive substances; GR: glutathione reductase; HUVECs: human umbilical vein endothelial cells.

Authors	Protocol	Patients	Target	Dose	Results
Caliskan [157]	Preclinical in vitro	-	Male Sprague-Dawley Rats	-	↓ MDA
Ishibashi [158]	Preclinical in vitro	-	HUVECs	-	↓ ROS production ↓ MCP-1 ↓ ICAM-1
Ishibashi [159]	Preclinical in vitro	-	Human proximal tubular cells	-	↓ ROS production ↓ MCP-1
Gul Utku [160]	Preclinical in vitro	-	Female Wistar rats	-	↓ MDA ↓ MPO
Ellinghaus [161]	Preclinical in vitro	-	HUVECs	-	↓ VCAM-1 ↓ ICAM-1, ↓ MCP-1 ↓ IL-8 ↓ CXCL1 ↓ CXCL2 ↓ TF
Vilaseca [162]	Preclinical in vitro	-	Wistar rats	-	↓ ROS
Maeda [163]	Preclinical in vitro	-	HUVECs	-	↓ ROS-induced senescence ↓ NOX subunits ↑ NOS
Shafiey [164]	Preclinical in vitro	-	Adult male Wistar rats	-	↑ SOD ↑ GP ↓ MDA ↓ NO ↓ NF-kB
Abedalqader [165]	Preclinical in vitro	-	Adult male Wistar rats	-	↓ TBARS
Imano [166]	Preclinical in vivo	-	Male C57BL/6J mice	-	↓ ROS ↓ NF-kB
Imam [167]	Preclinical in vivo	-	Adult male Wistar rats	-	↓ NOS ↑ GSH ↑ GR
Al-harbi [168]	Preclinical in vivo	-	Adult male Wistar rats	-	↓ MDA ↑ GSH ↑ GR
Abdelzaher [169]	Preclinical in vivo	-	Adult male Wistar rats	-	↓ MDA ↓ NF-Kb ↑ GSH ↑ SOD
Moñux [170]	Ex vivo	6	Abdominal aortic aneurysm sites with intraluminal mural thrombus	-	↓ NOX subunits ↓ NOS2 ↓ ICAM-1 ↓ VCAM-1

**Table 4 antioxidants-12-01185-t004:** **Main studies evaluating the antioxidant effects of apixaban**. ICAM-1: intercellular adhesion molecule 1; VCAM-1: vascular cell adhesion protein 1; ROS: reactive oxygen species; NOS: nitric oxide synthase; HUVECs: human umbilical vein endothelial cells; HMEC-1: human mammary epithelial cells.

Authors	Protocol	Patients	Target	Dose	Results
Torramade-Moix [173]	Preclinical in vitro	-	HUVECs and HMEC-1	-	↓ ICAM-1 ↓ VCAM-1 ↓ ROS ↑ NOS

**Table 5 antioxidants-12-01185-t005:** **Main studies evaluating the antioxidant effects of edoxaban**. ROS: reactive oxygen species; TNFa: tumor necrosis factor a; MCP-1: monocyte chemoattractant protein-1; NOX: NADPH oxidase; HK-2: human kidney 2.

Authors	Protocol	Patients	Target	Dose	Results
Narita [174]	Preclinical in vitro	-	HK-2 cells	-	↓ ROS
Fang [175]	Preclinical in vitro	-	Male wild-type mice and HK-2 cells	-	↓ TNFa ↓ MCP-1 ↓ NOX subunits

**Table 6 antioxidants-12-01185-t006:** **Main studies evaluating the antioxidant effects of dabigatran**. ROS: reactive oxygen species; MCP-1: monocyte chemoattractant protein-1; MDA: malondialdehyde; NF-kB: nuclear transcription factor—kB; TNFa: tumor necrosis factor a; CAT: catalase; SOD: superoxide dismutase; NOX: NADPH oxidase; GSH: reduced glutathione; iNOS: inducible nitric oxide synthase; TAC: total antioxidant capacity; TOS: total oxidant status; HUVECs: human umbilical vein endothelial cells.

Authors	Protocol	Patients	Target	Dose	Results
Kadoglou [176]	Preclinical in vitro	-	ApoE−/−mice	-	↓ ROS
Tripathy [178]	Preclinical in vitro	-	Rat brain endothelial cell cultures	-	↓ ROS ↓ MCP-1
Pingel [177]	Preclinical in vitro	-	ApoE−/−mice	-	↓ ROS
Yazici [181]	Preclinical in vitro	-	Male Sprague-Dawley rats	-	↓ MDA
Song [184]	Preclinical in vitro	-	Male New Zealand White rabbits	-	↓ NF-kB ↓ TNFa ↓ IL-1 ↑ CAT ↑ SOD
Wozńiak [183]	Preclinical in vitro	-	HUVECs	-	↓ ROS-induced DNA strand breakage ↓ ROS
Mahmoud [185]	Preclinical in vitro	-	Adult male albino rats	-	↓ MDA ↓ NOX ↓ TNFa ↓ IL-1 ↑ GSH
Iannucci [179]	Preclinical in vitro	-	Female transgenic Tg4510 AD mice	-	↓ iNOS ↓ NOX
Johnson [180]	Preclinical in vitro	-	LRRK2 mutant Drosophila melanogaster	-	↓ SOD ↓ NOX ↓ ROS
Ewees [187]	Preclinical in vitro	-	Male adult albino rats	-	↓ MDA ↑ GSH ↓ NOX ↓ SOD
Saifi [188]	Preclinical in vitro	-	Male Swiss albino mice	-	↓ ROS ↓ IL-1 ↓ TNFa ↑ GSH
Durmaz [182]	Preclinical in vitro	-	Male Wistar rats	-	↑ TAC ↑ TOS ↓ TNFa ↓ IL-1
Youssef [186]	Preclinical in vitro	-	Aimdualst female Wistar albino rats	-	↓ MDA

## Data Availability

Not applicable.

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
