# Peer review of "Antioxidant Properties of Oral Antithrombotic Therapies in Atherosclerotic Disease and Atrial Fibrillation"

_antioxidants, 2023, doi:10.3390/antiox12061185_

Round 1

Reviewer 1 Report

In their manuscript, the authors collected information on the broadly understood antioxidant effect of antiplatelet drugs and direct oral anticoagulants (DOACs).
The publication is important from a practical point of view, properly prepared, but I believe that a few threads should be expanded to cover the entire clinical problem:
- in relation to ticagrelor, there have been published works showing that ticagrelor, but not clopidogrel and prasugrel, prevents ADP-induced vascular smooth muscle cell contraction, moreover there is an interaction - high-dose, but not low-dose, aspirin impairs anticontractile effect of ticagrelor following ADP stimulation in rat tail artery smooth muscle cells. These interactions result from the specific extraplatelet distribution of P2Y12 receptors, which should be emphasized.
- in the case of DOAC treatment, there is a whole range of interactions with food - many of them are the result of the influence on the metabolic process - please mention the clinical significance of drug-food interactions of DOAC.
- interference in the oxidation processes may have both positive and negative effects from the clinical point of view. Please address the risk management of DOAC therapy in this situation. In general, DOAC testing should be considered in this situation. An example is - the interactions of nintedanib and oral anticoagulants > clinical implications. There are published long-term observational studies on risk management ie. therapeutic monitoring of direct oral anticoagulants - an 8-year observational study.
- at the end, I am asking for an extension summarizing the clinical implications - when, how to consider this type of therapy? can oxidative effects be of clinical significance and in whom? what kind of? whether to modify the therapy depending on these parameters?

Author Response

We thanks reviewer 1 for the proper suggestions that allow us to elucidate some aspects (modifications are in red in the updated manuscript):

  • Lines 322-326: In addition, Grzesk et al. [188] demonstrated that among P2Y12 inhibitors, only ticagrelor avoided the ADP-induced VSMC contraction. This effect, derived from interaction with extraplatelet-located P2Y12 receptors, enriches the evidence of pleiotropic effects of this new potent antiplatelet drug.
  • Lines 505-507: Even if much lower than VKAs, some drug-food and drug-drug interactions have been described during DOACs treatment due to the interaction with P450 cytochromes, ABC transporters, and P-glycoprotein (P-gp) [194 - 198].
  • Lines 511-517: The use of DOACs in daily clinical practice does not require monitoring of coagulation since all seminal RCTs comparing DOACs to VKAs have been conducted without dose adjustments based on plasma level measurements. However, conflicting results emerged from analysis of RCTs and long-term longitudinal studies. [199 – 202] Therefore, as-sessment of the anticoagulant effect of DOACs may be desirable in certain, rare situations such as extreme body weight, concomitant oncologic therapies, patients after trans-plantation, patients on HIV medication etc. [203 – 205]
  • Lines 489-495: The beneficial effect of aspirin, clopidogrel, ticagrelor and rivaroxaban on oxidative stress system has been showed in preliminary observational clinical studies including patients with CAD; however, the relationship between the antioxidant properties and the occurrence of cardiovascular events has not been demonstrated. The choice of the oral antithrombotic therapy based on the need of its antioxidant properties should follow a patient-centered approach. In the next future, the use of oxidative stress biomarkers could help to identify these patients. 

Reviewer 2 Report

Here are my comments:

1.     Title: Antioxidant proprieties of the oral antithrombotic therapies in atherosclerotic disease and atrial fibrillation

Proprieties => properties.

2.     Introduction: Oxidative stress plays a key role in the onset of vascular thrombosis, both arterial and venous [1].

Authors should introduce briefly what oxidative stress is in this part, and explain more specific in part 2: Role of oxidative stress in thrombosis.

3.     Introduction: Antithrombotic drugs have arguably deeply modified the natural history of thrombosis-related diseases; nevertheless, they continue to pose a tremendous burden on healthcare systems, suggesting the need for a deeper understanding of thrombotic mechanisms.

Why antithrombotic drugs pose a tremendous burden on healthcare systems?

4.     Introduction: However, most of ROS derive from mitochondria.

Add references.

5.     Role of oxidative stress in thrombosis - Role of Oxidative Stress in Venous Thrombosis

Add diagram to illustrate the role of oxidative stress in venous thrombosis.

6.     Role of oxidative stress in thrombosis: Atrial fibrillation (AF) is the major cause of cardioembolic stroke… the cardioembolic stroke is now the most frequent type of ischemic strokes.

Add references.

7.     Antioxidant effects of antiplatelet drugs: The potential antioxidative property of P2Y12 inhibitors has so far been evaluated mainly with Clopidogrel.

What is Clopidogrel? Authors should add a brief introduction about it.

8.     Antioxidant effects of antiplatelet drugs & Antioxidant effects of DOACs:

I would like to know the limitation & risk of each drug.

Moderate editing of English language

Author Response

We thanks reviewer 2 for the interesting suggestions that allow us to clarify important aspects of the manuscript (modifications are in red in the updated manuscript).

1) Line 1. Title corrected.

2) As suggested a briefly introduction on oxidative stress has been added in the introduction while more detailed information on the relation between oxidative stress and thrombosis have been included.

Lines 34-38: Oxidative stress occurs when there is a discrepancy between the rate of reactive oxygen species (ROS) generation and elimination. Normally, they are produced as waste products of oxygen metabolism, but environmental factors can trigger a dramatic increase in ROS production, tipping the scales toward an unbalanced state that results in cellular and tissue damage.[1] 

Lines 121-140: 
Oxidative stress and coagulation are closely intertwined. Indeed, ROS work at different levels in the coagulation landscape, involving endothelium, platelets, and coagulation factors. [178] One of the first consequences of an increased ROS production is endothelial dysfunction (ED). The hallmark of ED is the shift from a normal endothelium phenotype that hampers platelets activation and foster vasodilatation, towards an endothelium promoting a procoagulant state. [179] Additionally, platelets furtherly boost oxidative stress, establishing a vicious circle that can hasten and strengthen the thrombotic process. [178] Mechanistic studies investigating molecular pathways underlying ROS effects on platelets function are lacking. However, it has been hypothesized that function of crucial receptors may be positively modulated by oxidative changes on sulfhydryl and thiol groups. [180] Therefore, coagulation factors generation and receptor-binding are boosted, and platelet adhesion is enhanced. Red blood cells (RBCs) are affected too. RBCs, once thought to be passive spectators of hemostasis, are now recognized as crucial agents in fostering venous thrombosis and improving thrombus stability. [68, 181] Oxidative stress can disrupt membrane structure and foster phosphatidylserine exposition, laying the groundwork for prothrombin cleavage. [182] In addition, RBCs mechanical properties are impaired, thus blood viscosity is enhanced. Subsequently, local blood flow is lowered, enabling RBCs aggregation, and supporting platelet adhesion. [183]      

3) The sentence has been corrected.
Lines 55-58: Antithrombotic drugs have deeply modified the natural history of thrombosis-related diseases. Nevertheless, thrombotic disorders continue to pose a tremendous burden on healthcare systems, suggesting the need for a deeper understanding of thrombotic mechanisms.

4) Reference added.
Lines 112-113: However, most of ROS derive from mitochondria. [177]

5) Diagram added.
Lines 194-197: Figure 2 added. See updated manuscript.

6) Reference added.
Line 202: 
stroke is now the most frequent type of ischemic strokes. [8]

7) A briefly introduction of clopidogrel has been included.
Lines 311-318: 
Clopidogrel is a second generation thienopyridine that binds irreversibly P2Y12 inhibiting platelet aggregation. As an inactive prodrug, clopidogrel needs to undergo hepatic bioactivation. However, this step involves only a 15% of administered prodrug, as the remaining 85% is extensively converted into a non-functional metabolite. [185] Despite the pharmacokinetic profile and the rise of more potent third generation P2Y12 inhibitors (prasugrel and ticagrelor), clopidogrel is still widely prescribed. [186,187]

Moreover a brief introduction on P2Y12 has been also added.
Lines 306-310: 
P2Y12 activation by ADP is essential for thrombus formation. This interaction leads to the release of mediators contained in dense granules and to inhibition of intracellular prostacyclin pathway, thereby further promoting platelet aggregation. Thus, P2Y12 has become an attractive target for modern antiplatelet therapies. [184]

8) Risk and limitations of both DOACs and antiplatelet drugs have been emphasized. As reviewer 1 also suggested a focus on specific DOACs limitations (in particular drug-food interactions and therapeutic monitoring) we suggest a proper paragraph illustrating the significant constraints of these medications that clinicians face in everyday clinical practice. 

Lines 534-572: 

5. Clinical Implications

5.1. Current challenges in everyday clinical practice

Significant advances in antithrombotic therapy have been accomplished over the last years. At the same time, long-term management of antiplatelet and anticoagulant regimens has become increasingly complex. In everyday clinical practice clinicians face multiple challenges. Bleeding risk is a main issue, as clinicians have to carefully monitor variations of this risk over time and take into account comorbidities and residual thrombotic risk. Additionally, intolerances or adverse effects such as ticagrelor-induced dyspnea shall be monitored. The introduction of novel P2Y12 inhibitors further reduced ischemic events at the price of an increased rate of bleedings. Therefore, several strategies have been suggested as potential replacements for traditional double antiplatelet therapy (DAPT). P2Y12 inhibitor or aspirin monotherapy after a short period of DAPT or early shift from newer P2Y12 inhibitors to clopidogrel are promising strategies to blunt the bleeding risk. [189,190] Despite a more favorable bleeding profile compared to VKAs, DOACs still have limitations. Indeed, DOACs are not immune to adverse drug-food and drug-drug interactions (DFI and DDI respectively). [191, 192] Most of macronutrients do not affect DOACs bioavailability. A high intake of fibers, though, can reduce DOACs plasmatic concentrations. Additionally, caution is needed when taking dietary supplements. In fact, these compounds affect P450 cytochromes and ABC transporters function, leading to either an increase or a decrease in DOACs concentration. Therefore, an enhanced bleeding risk or a lower protection against thrombosis respectively, occurs. Despite significant differences in pharmacokinetic profile, all DOACs share the interaction with P-glycoprotein (P-gp), while rivaroxaban and apixaban concentrations are also influenced by several drugs disrupting P450 cytochromes activities. As patients taking DOACs are gradually getting elderly and comorbid, polypharmacy has become a crucial issue. A myriad of different drugs impacts on DOACs bioavailability, from commonly used antiarrhythmic drugs, to medications addressing rare diseases, such as nintedanib. [192,193] Therefore, clinicians’ awareness is critical to reduce adverse events and improve patients’ outcomes. A potential solution is represented by therapeutic drug monitoring (TDM) of DOACs. Although DOACs are characterized by predictable pharmacokinetic profile, residual thrombotic and bleeding risk may be attributed to inappropriate DOACs levels. On one hand there is evidence that DOACs concentration is related to adverse events. [194,195] On the other hand, since data from both longitudinal studies and RCTs failed to show a link between plasmatic levels and outcomes, [196,197] TDM should be limited to high-risk situations.

Round 2

Reviewer 1 Report

the work has been significantly enriched. I believe it may be considered for publication

Reviewer 2 Report

NO COMMENTS

NO COMMENTS